# The Role of P-Selectin in COVID-19 Coagulopathy: An Updated Review

**DOI:** 10.3390/ijms22157942

**Published:** 2021-07-26

**Authors:** Chiara Agrati, Alessandra Sacchi, Eleonora Tartaglia, Alessandra Vergori, Roberta Gagliardini, Alessandra Scarabello, Michele Bibas

**Affiliations:** National Institute for Infectious Diseases “Lazzaro Spallanzani” I.R.C.C.S., Via Portuense 292, 00146 Rome, Italy; chiara.agrati@inmi.it (C.A.); alessandra.sacchi@inmi.it (A.S.); eleonora.tartaglia@inmi.it (E.T.); alessandra.vergori@inmi.it (A.V.); roberta.gagliardini@inmi.it (R.G.); alessandra.scarabello@inmi.it (A.S.)

**Keywords:** P-selectin, COVID-19, SARS-CoV-2

## Abstract

In severe COVID-19, which is characterized by blood clots and neutrophil-platelet aggregates in the circulating blood and different tissues, an increased incidence of cardiovascular complications and venous thrombotic events has been reported. The inflammatory storm that characterizes severe infections may act as a driver capable of profoundly disrupting the complex interplay between platelets, endothelium, and leukocytes, thus contributing to the definition of COVID-19-associated coagulopathy. In this frame, P-selectin represents a key molecule expressed on endothelial cells and on activated platelets, and contributes to endothelial activation, leucocyte recruitment, rolling, and tissue migration. Briefly, we describe the current state of knowledge about P-selectin involvement in COVID-19 pathogenesis, its possible use as a severity marker and as a target for host-directed therapeutic intervention.

## 1. Introduction

The severe acute respiratory syndrome coronavirus 2 (SARS-CoV-2) causes coronavirus disease 2019 (COVID-19), which can present with a wide range of manifestations, including pneumonia and acute respiratory distress syndrome (ARDS) [1]. COVID-19 has a very different clinical spectrum, ranging from asymptomatic or paucisymptomatic (in up to 45 percent of people) to severe diseases necessitating intensive care unit admission (ICU). As seen in patients with COVID-19 and post-mortem samples from those who died after SARS-CoV-2 infection, SARS-CoV-2 infection causes a pro-thrombotic condition manifested mainly by microthrombosis [2]. Excessive pulmonary immunothrombosis, an intrinsic route of innate immunity induced by pathogens and wounded cells to prevent the spread and survival of invading pathogens, is connected to patients with severe COVID-19 who have an extremely high risk of thrombosis [3]. Excessive stimulation of the immunothrombosis process causes thromboinflammation, a condition in which inflammation and thrombosis coexist within the microvessels in response to damaging stimuli. Immunothrombosis is defined by the production of microthrombi in tiny capillaries, in which endothelial cells exposed to microorganisms adopt a pro-adhesive phenotype. Neutrophils and monocytes are primarily responsible for this process [4]. The fast-paced nature of this discipline needs the integration of existing biology data with COVID-19 clinical findings in order to better understand the disease’s etiology and contribute to the development of new possible therapeutics. In this review, we summarize the current published studies that show the role of P-selectin in COVID-19 associated coagulopathy.

## 2. Selectins

The selectins are a family of calcium-dependent (C-type) lectins best known for mediating immune cell adherence to the endothelium, allowing immune cells to enter secondary lymphoid organs and inflammatory sites. The selectin family consists of three members, each called for their expression patterns: P-selectin, E-selectin, and L-selectin, which are expressed on platelets, endothelial cells, and leukocytes, respectively [5]. Only P-selectin was the subject of our investigation.

P-selectin is a type-1 transmembrane protein encoded by the SELP gene in humans. P-selectin is found on chromosome 1q21-q24, spans > 50 kb, and has 17 exons. Megakaryocytes (platelet precursors) and endothelial cells express P-selectin on a constant basis [6]. Two different processes are involved in the induction of P-selectin expression. First, megakaryocytes and endothelial cells produce P-selectin, which is then deposited in platelet alpha granules and endothelial cell Weibel-Palade bodies. When a cell is exposed to an activating stimulus like thrombin, P-selectin is rapidly translocated to the cell surface, eliminating the requirement for transcription or translation. Within minutes after activation, P-selectin is transported to the exterior plasma membrane. This increase in P-selectin expression is very temporary, as the protein is quickly internalized and destroyed or recycled within the cell. TNF also increases the transcription of P-selectin [6]. P-selectin binds to heparan sulfate and fucoidans, but its principal ligand is P-selectin glycoprotein ligand-1 (PSGL-1), which is expressed on practically all leukocytes. PSGL-1 is found on a variety of hematological cells, including neutrophils, eosinophils, lymphocytes, and monocytes, where it promotes cell tethering and adhesion [7]. P-selectin is involved in the initial attachment and rolling of platelets and leukocytes to inflamed and injured regions. PSGL-1 signaling in leukocytes and platelets, as well as GPIb signaling in platelets, play important roles in hemostasis and thrombosis. The VWF receptor, glycoprotein (GP) Iba, has also been demonstrated to be a counter-receptor for P-selectin, suggesting that platelet rolling on P-selectin may be facilitated [8]. Platelet and leukocyte activity, in this case, are strikingly similar, both cell types must slow down (roll) before firmly adhering to the site of injury/inflammation. The fact that the adhesion molecules that cause initial adhesion in both leukocytes and platelets are kept in the same organelle and are thus always released jointly demonstrates how closely hemostatic and inflammatory responses are linked [9]. Furthermore, soluble P-selectin (sP-sel) is detected in the circulation as a monomer, indicating that sP-sel must dimerize to activate signaling in leukocytes, according to in vitro investigations. As a result, when sP-sel dimerizes or is identified as a dimer on the surface of platelet-derived microparticles, it can activate leukocytes, making it not only a helpful biomarker but also a direct contributor to vascular disease [10].

## 3. P-Selectin in Human Diseases

P-selectin levels in the plasma have been found to be elevated in a number of human illnesses. Endothelial cells in Sickle Cell Disease (SCD) express P-selectin on a long-term basis [11]. P-selectin upregulation in endothelial cells and platelets contributes to cell–cell interactions implicated in the development of vaso-occlusion and sickle cell pain crises [12]. P-selectin levels in the blood have been found to be elevated in a variety of acute and chronic cardiovascular diseases, including peripheral arterial disease, coronary artery disease, hypertension, and acute myocardial infarction [13,14,15,16,17]. P-selectin is also involved in atherosclerosis [18], as well as hypercholesterolemia [19,20]. In patients with symptomatic internal carotid artery stenosis, elevated levels were found, with further increases in acute ischemic stroke [21]. When compared to healthy controls, P-selectin levels were found to be 2- to 2.5-fold higher in patients on the first day following cerebral ischemia. P-selectin levels continued to drop steadily from the second day onwards, eventually returning to normal 90 days following the stroke [22]. Increased prothrombotic activity, impaired fibrinolysis, diminished endothelial thromboresistance, and platelet hyperreactivity with overall hypercoagulability are common in diabetic patients. Many investigations have found that patients with diabetes mellitus have a greater median plasma level of circulating P-selectin, as well as enhanced P-selectin expression on platelets [23]. This activation, it should be noted, was not linked to improved glycemic control with increased insulin therapy [24]. P-selectin may be responsible for increased platelet activation in deep venous thrombosis (DVT), in addition to its effect on arterial thrombosis. Furthermore, it may play a role in the diagnosis of DVT, according to a meta-analysis report that included eleven trials [25]. P-selectin may play a predictive role in the diagnosis of DVT in cancer patients, according to other researchers [26]. In venous thrombosis, higher levels of sP-sel were discovered than in advanced-stage cancer patients, leading to the hypothesis that, despite an increase in sP-sel in patients with metastatic cancer, their values rise in acute DVT [27]. Thrombotic consumptive diseases, such as disseminated intravascular coagulation, thrombotic thrombocytopenic purpura, and heparin-induced thrombocytopenia, also have higher levels of P-selectin [28,29,30]. Oncology is one area where P-selectin’s role has been demonstrated convincingly in both experimental and clinical research. P-selectin is one of the most well-studied platelet-tumor interaction mediators. Heparin inhibits tumor cell spread in vivo by inhibiting the P-selectin-mediated platelet-tumor interaction [31]. One possible explanation is that P-selectin is required for the creation of the platelet cloak that surrounds circulating tumor cells and protects tumor cells from NK cell attack [32]. While P- and L-selectins have been found to have a synergistic effect on tumor cell systemic dissemination in vivo, heparin’s inhibitory influence on this process is related to the blockage of P-selectin function [33]. Furthermore, P-selectin’s cytoplasmic domain appears to facilitate platelet infiltration into tumors by binding to talin-1, inducing talin1-mediated activation of aIIbb3 integrin and hence platelet recruitment into tumors [34]. Previous research has found that enhanced P-selectin expression occurs during viral infections, including influenza [35]. Increased P-selectin expression enhances platelet-leukocyte aggregation formation via P-selectin glycoprotein ligand-1 (PSGL-1). Platelet-leukocyte aggregates (PLA) production has been described as a highly sensitive measure of platelet activation in vivo during viral illnesses [36]. Several investigations have found that patients living with HIV have higher levels of P-selectin than people who are not infected. This backs up the observation that adult HIV-infected patients have higher levels of platelet activation [37]. Importantly, these levels were found to persist in follow-up investigations after 3 to 24 months of effective ART [38]. Despite the fact that ART reduced platelet activation in individuals who did not receive protease inhibitors (PIs), platelet activation levels in those on PI-based therapy persisted despite successful ART [39,40,41]. High plasma soluble P-selectin concentrations at baseline were connected to reduced FEV1 and RV/TLC, a sign of severe air trapping in asthma patients [42]. Severe obstructive sleep apnea was also linked to higher plasma P-selectin levels, which were correlated to illness severity measures such as the apnea-hypopnea index, oxygen desaturation index, and respiratory disturbance index [43,44].

## 4. Methodology

To give the most thorough background information on P-selectin in COVID-19, we did an exhaustive literature search utilizing PubMed/Medline, EMBASE, and Google Scholar to find publications published in English between January 2020 and June 2021. The main search terms used were “P-selectin” or “sP-selectin” and “COVID-19” or “SARS-CoV-2”

## 5. Search Strategy

The search procedure used in this systematic review is visualized in Figure 1. All publications were reviewed by two separate reviewers (CA and MB) on the basis of title, abstract, and full-text level. The systematic review covered all papers on the function of P-selectin in the pathophysiology of COVID-19. Titles and abstracts chosen by one of the reviewers were included in the full-text screening process. Final articles that matched our study topic were withheld after rigorous reading and review of each selected full-text. Both independent reviewers also searched the reference lists of the selected full-text articles for supplemental papers. The same two reviewers went through the same selection process for these new articles. Articles written in languages other than English were not included in the study. All study designs were considered, with the exception of editorials, single-case reports, and reviews. Observational studies were mostly chosen.

## 6. Results

Our initial search rendered 56 articles. After reviewing the full-text articles, 20 studies were selected, and data from 936 cases were analyzed. (Table 1)

Five studies have associated P-selectin levels with critical illness and death [45,46,47,48,49]. Goshua et al. [45] found P-selectin and other platelet and endothelial markers significantly elevated in intensive care unit (ICU) patients compared with controls, and also significantly higher in ICU patients than in non-ICU patients. Mortality was significantly correlated with the elevation of those markers. Hottz et al. [46], evaluating the P-selectin values within 72 h from ICU admission, demonstrated an increased level of those patients compared with healthy control and asymptomatic or mildly ill infected patients. Patients’ poor outcomes, including the need for mechanical ventilation and in-hospital mortality, were predicted by P-selectin levels above the control group median and platelet-dependent TF expression in monocytes upon admission. Patients with severe COVID-19 syndrome had increased platelet activation and platelet-monocyte aggregation formation, however, patients with mild self-limiting COVID-19 disease did not. Although P-selectin was the predominant adhesion molecule enabling platelet-monocyte aggregation formation in this scenario, platelet-induced TF expression needed both P-selectin and integrin aIIb/b3 signaling. Pretreatment with aspirin and clopidogrel did not inhibit platelet-induced TF in monocytes, which was interesting. After adjusting for confounding factors, Campo et al. [47], discovered that P-selectin values were greater and had a different pattern over time in patients who died versus those who survived. P-selectin levels that were higher were likewise linked to myocardial damage. This study backs up previous findings that COVID-19 is linked to changes in platelet activation and aggregation, and that there are no variations in platelet aggregation values between ICU and non-ICU COVID-19 patients. P-selectin and sCD40L values, on the other hand, were influenced by COVID-19 severity, with P-selectin and sCD40L levels being higher in ICU patients.

In order to predict death, Vassiliou et al. [48] looked at patterns of several endothelium-related indicators in critically ill COVID-19 patients on ICU admission. COVID-19 critically sick patients who would not survive had higher ICU entry levels of sP-sel and other endothelial markers. Non-survivors had a cumulative theoretical predictive score of 4.1, compared to 1.4 for survivors, based on biomarkers. They did not find greater levels of sP-sel in severely ill COVID-19 patients in this cohort when compared to patients hospitalized in the ward. This is most likely related to the fact that markers were tested on admission rather than during hospitalization. Barrett et al. [49], following adjustment for age, sex, race/ethnicity, antiplatelet therapy, platelet count, and chronic obstructive pulmonary disease, TxB2, P-selectin, sCD40L were independently associated with the risk of thrombosis or death.

Two studies reported no difference in P-selectin levels in ICU and non-ICU groups [50,51]. Compared to normal reference values and contextually sampled healthy donors, Agrati et al. [50] found a greater P-selectin plasma concentration in patients with COVID-19, regardless of ICU admission. Furthermore, after platelet removal in HD, data revealed a large reduction in P-selectin, implying that the majority of this molecule was stuck in the platelets. In contrast, both ICU and non-ICU COVID-19 patients had equal amounts of P-selectin with and without platelets, implying that COVID-19 caused these molecules to be released from active platelets/cells. In both groups, platelet counts were found to be comparable. ICU patients had a considerably lower lymphocyte count, demonstrating the link between lymphocytopenia and illness severity. P-selectin was found to be considerably higher in all COVID-19 patients compared to healthy donors, according to Manne et al. [51]. In terms of P-selectin expression, there was no difference between ICU and non-ICU COVID-19 patients. Surprisingly, mRNA from the SARS-CoV-2 N1 gene was found in platelets from two of the 25 COVID-19 patients, suggesting that platelets may take up SARS-COV-2 mRNA without requiring ACE2. P-selectin expression was higher in COVID-19 patients’ resting platelets and when they were activated. COVID-19 patients had considerably higher circulating platelet-neutrophil, -monocyte, and -T-cell aggregates than healthy donors. Platelets from COVID-19 patients also aggregated more quickly and distributed more widely by fibrinogen and collagen. [51] Five studies found higher values in COVID-19 ICU patients compared with severe non-ICU COVID-19 patients [52,53,54,55,56]. SARS-CoV-2 interacts with platelets and megakaryocytes via an ACE2-independent mechanism, according to Shen et al. [52], and may influence alternative receptor expression linked with COVID-19 coagulation abnormalities. Increased P-selectin translocation on platelet surfaces revealed a direct contact between SARS-CoV-2 and human platelets. A higher value of P-selectin in ICU patients compared with severe COVID-19 non-ICU patients was reported. [52]

In individuals with COVID-19, circulating platelets and neutrophils are highly stimulated, according to Petito et al. [53]. Patients with COVID-19 have considerably higher levels of soluble P-selectin and neutrophil-derived microparticles (PMN-MPs), suggesting that they could be used as simple platelet and neutrophil activation indicators in SARS-CoV-2 infection. Even though the majority of the cases were modest, they discovered significant platelet and neutrophil activity. Platelet and neutrophil activation indicators, on the other hand, were linked to the severity of COVID-19. Furthermore, platelet and neutrophil activity returned to normal in COVID-19 individuals. Plasma from COVID-19 patients activated platelets and neutrophils, resulting in neutrophil extracellular traps (NETs) formation, implying that inflammatory mediators, most likely cytokines, generated during the SARS-CoV-2 infection are responsible for the in vivo platelet and neutrophil activation seen in COVID-19 patients [53].

The hyperactive platelet pattern was verified by Comer et al. [54]. When compared to controls, the circulating levels of the platelet activation markers PF4 and sP-sel were significantly higher in COVID-19–positive individuals. While circulating levels of PF4 did not differ between severe and non-severe COVID-19 individuals, the severe COVID-19 group had greater sP-sel. Fraser et al. [55] used a machine learning system for thrombosis profiling (using P-selectin value) in ICU, predicting severity and mortality of COVID-19. Three thrombotic factors and five endothelial cell damage markers were evaluated in plasma from COVID19^+^ and COVID19^–^ ICU patients, as well as age and sex-matched healthy control subjects. COVID-19^+^ patients had higher vWF than healthy control participants, but more crucially, in the plasma of COVID-19^+^ patients, sP-sel was considerably raised by ICU day 3 and remained persistently elevated in plasma until ICU day 7. COVID-19^+^ patients had exacerbated and chronic endothelium damage, as evidenced by raised sP-sel. Karsli et al. [56] focused on the diagnostic and predictive relevance of serum soluble sP-Sel levels in COVID-19 illness and found that patients with mild-to-moderate and severe pneumonia had greater serum sP-Sel levels than those with no pneumonia. At the cut-off level of 4.125 ng/ml, sP-Sel levels were reported to be 97.5 percent sensitive and 80 percent specific in the diagnosis of COVID-19. Patients with an inflammatory reaction had higher sP-Sel levels than the control group, and their risk of endotheliopathy and thrombosis was higher.

Four studies were only able to report higher values of P-selectin in COVID-19 patients than healthy controls, confirming the pro-thrombotic platelet phenotype, in SARS-CoV-2 infected patients [57,58,59,60]. Taus et al. [57], have addressed COVI-19 induced changes in platelet subpopulations as well as exploring the role of platelet in driving thromboinflammation through the release of inflammatory cytokines. After carefully selecting patients without concurrent conditions known to affect platelet function, the study found that patients with COVID-19 had increased basal expression of P-selectin on platelets, which was unaltered upon further agonist stimulation, suggesting a procoagulant platelet status that could not be enhanced. Using paired analysis, Chao et al. [58], evaluated platelet and leukocyte activation in consecutive samples from a patient cohort throughout the acute and convalescent phases of COVID-19. The baseline CD62P surface expression levels in this sample of primarily mild and moderate COVID-19 patients were similar in the acute and convalescent phases. During the acute phase, platelet populations were much more susceptible to conventional platelet agonists, ADP and thrombin, than during the convalescent phase. According to the researchers, platelet activation was an early response to mild or moderate COVID-19, and it was not exclusively associated with a severe disease.

Canzano et al. [59], found a substantial increase in platelet P-selectin expression (10-fold higher than in healthy subjects). Surprisingly, they discovered a negative relationship between circulating PLA and levels of IL-6, CRP, and D-dimer.

Bongiovanni et al. [60] also studied the expression of activation indicators and transmembrane receptors in platelets from hospitalized stable COVID-19 patients who did not have any pre-existing diseases and were not on any anticoagulants or antiplatelet drugs (except prophylactic low-molecular-weight heparin during hospitalization). They limited the measurements to stable COVID-19 patients who did not require assisted ventilation or extracorporeal perfusion, which could cause platelet activity unrelated to the condition. In comparison to controls, they found significantly greater levels of the platelet activation marker P-Selectin. Furthermore, compared to controls, COVID-19 patients had faster platelet aggregation and greater spreading of fibrinogen and collagen.

COVID-19 patients have a lower P-selectin concentration than control samples, according to Venter et al. [61]. However, compared to the results of other investigations, the concentrations of sP-sel in controls were substantially greater. The platelet physiology disruption in COVID-19 individuals was profound. COVID-19 samples were substantially more hypercoagulable and viscous than healthy controls, according to TEG^®^ and viscometry data.

In a retrospective cohort analysis of 79 hospitalized COVID-19 patients, Clark et al. [62] looked at platelet reactivity. They were unable to add healthy controls for comparison during the early stages of the pandemic, which hampered this study. As an alternative, they included hospitalized individuals who tested negative for COVID-19 but were diagnosed with something else. Based on P-selectin activation, there was no difference in platelet reactivity between the COVID-19 positive and negative control groups of hospitalized patients.

Patients hospitalized with non-severe COVID-19 have lower platelet reactivity than healthy controls, according to Bertolin et al. [63]. The findings of platelet hyporesponsiveness could be attributed to platelets being significantly stimulated in vivo during COVID-19, leading to subsequent refractoriness to new agonists administered during ex vivo platelet function testing, according to the author. There was no difference in P-selectin levels between non-severe COVID-19 patients and healthy controls. In a single-center prospective analysis, Spadaro et al. [64] sought to describe significant differences between COVID-19-related and traditional ARDS. Because they only involved patients who required mechanical ventilation, the findings could not be applied to mild or moderate COVID-19. P-selectin levels were greater in classical ARDS patients than in COVID-19 ARDS patients in those patients. The most severe forms of COVID-19 ARDS are characterized by the predominance of “endothelial” injury over “alveolar” injury, as evidenced by higher levels of Ang-2 and ICAM-1 in non-survivors compared to survivors. COVID-19 ARDS and classical ARDS both had a similar loss in gas exchange but different biomarker expression, suggesting different pathological pathways.

## 7. Discussion

The existing literature on the role of P-selectin in COVID-19 patients suggests that it could be a valuable biomarker for predicting clinical outcomes in COVID-19 patients. However, the paucity and inconsistency of existing evidence, the absence of standardized methodologies for biomarker testing and assessment, and the lack of prospective validation in relevant patient populations all may restrict the clinical relevance of P-selectin as a biomarker. The pathophysiological mechanism underlying COVID-19 patients’ increased rate of thromboembolic events is unknown. Upregulation of P-selectin on injured endothelial cells and activated platelets, on the other hand, contributes to a pro-thrombotic state that leads to immunothrombosis and thromboinflammation.

Based on our current understanding of the processes of thrombocytopathy and endotheliopathy, which are highlighted in this P-selectin review, drugs targeting platelet activation and death, as well as increasing endothelial cell health, could be useful in treating COVID-19 patients. Although aspirin is only suggested in COVID-19 antithrombosis reviews as a treatment for acute arterial thrombotic complications, prophylactic antiplatelet agents are now being included in therapeutic algorithms for the management of COVID-19 patients, and many clinical trials on the prophylactic use of antiplatelet agents have been proposed or are currently underway.

Dipyridamole (DIP) is an antiplatelet drug that works by inhibiting phosphodiesterase (PDE) and increasing intracellular cAMP/cGMP levels. DIP supplementation was related to significantly lower D-dimer concentrations, greater lymphocyte, and platelet recovery in the circulation, and markedly improved clinical outcomes in a proof-of-concept experiment including 31 patients with COVID-19 compared to control patients [65]. Dipyridamole’s effectiveness in hospitalized COVID-19 patients is being studied in randomized clinical studies. Three small randomized clinical trials are evaluating dipyridamole 100 mg four times a day and a combination of dipyridamole extended-release 200 mg twice daily and aspirin 25 mg twice daily. (TOLD, ClinicalTrials.gov Identifier: NCT04424901; DICER ClinicalTrials.gov Identifier: NCT04391179; and ATTAC-19 ClinicalTrials.gov Identifier: NCT04410328). The primary objectives included reductions in D-dimer levels (for the first two trials) and improvements in the COVID-19 WHO ordinal scale (a scale that ranks the severity of sickness from 0 [not infected] to 8 [death])(ATTAC-19).

Aspirin is an inexpensive, widely available treatment that inhibits the COX-1 enzyme, which is essential for the formation of thromboxane A2 and pro-inflammatory prostaglandins, irreversibly at low dosages. Aspirin has been demonstrated to prevent both arterial and venous thrombotic events in SARS-CoV-2 infected individuals and to abolish in-vitro hyperactivity in platelets [51,66]. While the effects of aspirin on clinical outcomes of community-acquired pneumonia patients have been investigated, providing preliminary results on its potential usefulness in lowering mortality, data regarding the efficacy of antiplatelet drugs in COVID-19 is scarce. Chow et al. [67] studied 420 COVID-19 patients. Of these, 314 (76.3%) were aspirin-free and 98 (23.7%) were on aspirin within 24 h of admission or 7 days prior to admission. After adjusting for eight confounding variables, aspirin use was found to be associated with a 46% lower risk of mechanical ventilation, a 43% lower risk of ICU admission, and a 47% lower risk of in-hospital mortality, with no differences in major bleeding thrombosis between aspirin users and non-users [67]. The small sample study and the retrospective nature of the study do not allow for definite conclusions. Further, allocation to aspirin was not associated with reduced mortality or, among those not on invasive mechanical ventilation at baseline, the risk of progressing to the composite endpoint of invasive mechanical ventilation or death in the only large published (medRxiv) randomized trial (RECOVERY) [68], which included over 14,000 patients and over 2000 deaths. However, being assigned to aspirin was linked to a slight increase in the chance of being discharged alive from the hospital after 28 days. As expected, aspirin treatment was linked to a higher risk of severe bleeding and a lower risk of thromboembolic consequences, with roughly six more patients experiencing a major bleeding event and six fewer experiencing a thromboembolic event for every 1000 patients treated with aspirin [68]. Furthermore, any potential benefit of antithrombotic treatments in COVID-19 patients may be contingent on treatment initiation timing, particularly if thrombi have already been formed by the time of admission. The apparent absence of effect in the INSPIRATION and REMAP-CAP/ACTIV-4a/ATTACC severe illness cohorts, in this opinion, shows that these patients may have passed the phase when therapeutic anticoagulation could be beneficial [69,70]. Inclacumab, which is in Phase 3, and Crizanlizumab, which has been approved by the FDA and EMA, are anti-P-selectin monoclonal antibodies that have been developed for human use. The latter, a humanized IgG2 kappa monoclonal antibody that blocks leucocyte and platelet adhesion to the artery wall, is used to avoid vaso-occlusive crises and reduce hyperinflammation in adults and juveniles patients with sickle cell disease [71]. A clinical trial (CRITICAL ClinicalTrials.gov Identifier: NCT04435184) to assess the efficacy and safety of Crizanlizumab in hospitalized adult patients with moderate COVID-19 was just completed and results are pending.

## 8. Conclusions

Due to the increased risk of morbid sequelae in COVID-19 patients, more large-scale prospective trials to examine the usefulness of P-selectin as a platelet and endothelial activation marker, to stratify risk, and for unfavorable prognostic outcomes are urgently needed. This type of study offers the potential to disclose the mechanisms driving platelet activation and endothelial injury, as well as to identify specific treatments aimed at minimizing endothelial activation and lowering the risk of thrombosis.

## Figures and Tables

**Figure 1 ijms-22-07942-f001:**
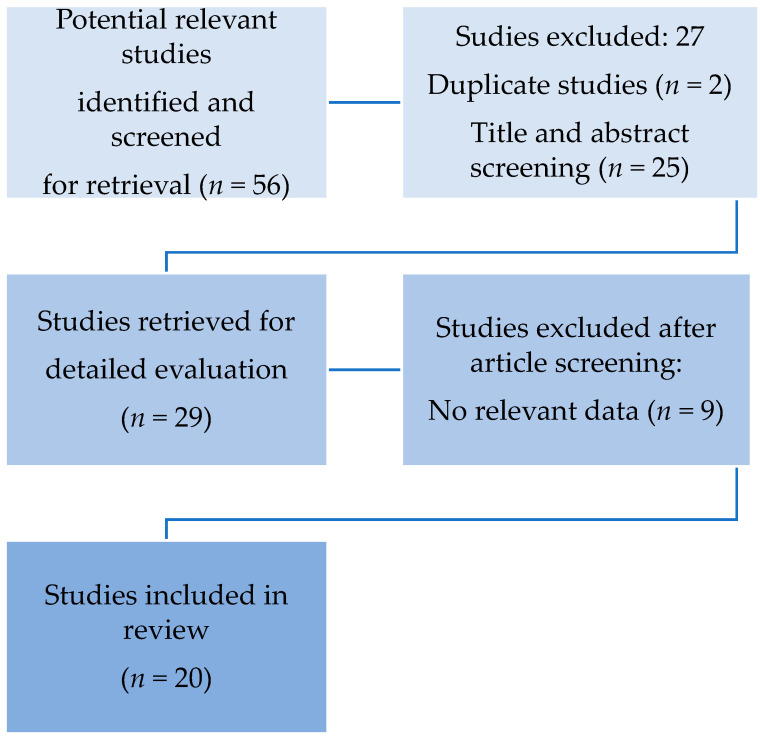
Search strategy.

**Table 1 ijms-22-07942-t001:** Studies on P-selectin in COVID-19 patients.

AUTHOR	COVID-19 pts	Age Median (Range)	Gender M/F	Moderate COVID-19+ (n)	Severe non- ICU COVID-19+ (n)	Severe ICU COVID-19+ (n)	Main Findings
Goshua [45]	68	62 (20–93)	27/41	-	higher than HC (*n* = 20)	higher than non-ICU (*n* = 48)	Association with critical illness and death
Hottz [46]	41	57 (47–64)	19/22	-	-	higher than controls (*n* = 35)	Association with critical illness and death
Campo [47]	54	65 (57–73)	40/14	-	higher than HC (*n* = 17)	higher than non-ICU (*n* = 37)	Association with critical illness and death
Vassiliou [48]	38	65(51–78)	31/7	-		higher than non-ICU (*n* = 38)	Association with critical illness and death
Barrett [49]	100	65	61/39		higher than HC (*n* = 100)		Association with critical illness and death
Agrati [50]	46	67 (50–80)	34/12	-	higher than HC (*n* = 19)	no differences with non-ICU (*n* = 27)	Higher P-Sel regardless ICU admission
Manne [51]	41	55 (33–77)	19/22	-	higher than HC (*n* = 24)	no differences with non-ICU (*n* = 17)	Higher P-Sel regardless ICU admission
Shen [52]	62	66 (60–71)	32/30	-	higher than HC (*n* = 37)	higher than non-ICU (*n* = 25)	Higher values in COVID-19+ ICU patients.
Petito [53]	36	70.6 (36–60)	20/16	-	higher than HC (*n* = 17)	higher than non-ICU (*n* = 19)	Higher values in COVID-19+ ICU patients
Comer [54]	54	63 (47–84)	34/20	-	higher than HC (*n* = 20)	higher than non-ICU (*n* = 34)	Higher values in COVID-19+ ICU patients
Fraser [55]	10	61 (54.8–67)	3/7	-	-	higher than HC (*n* = 10)	Higher values in COVID-19+ ICU patients
Karsli [56]	80	-	-		higher than HC (*n* = 50)	higher than non-ICU (*n* = 35)	Higher values in COVID-19+ ICU patients
Taus [57]	37	61.8 (47–94)	18/19	-	higher than HC (*n* = 37)	-	P-Sel higher than controls
Chao [58]	15	71 (53–80)	5/10	higher than HC (*n* = 5)	-	-	P-Sel higher than controls
Canzano [59]	46	72 (58–84)	28/18	-	higher than HC (*n* = 46)	-	P-Sel higher than controls
Bongiovanni [60]	8	51.4 (39–64)	5/3	-	higher than HC (*n* = 8)	-	P-Sel higher than controls
Venter [61]	30	53.1 (38–69)	-	lower than HC (*n* = 30)	-	-	Study reporting a lower P-Sel level in COVID-19+
Clark [62]	79	67 (54–75)	43/34	-	Similar to Controls (*n* = 79)	-	Study reporting a similar P-Sel level In COVID-19+ and Controls
Bertolin [63]	60	52 (37–68)	31/29	Similar to Controls (*n* = 60)			Study reporting a similar P-Sel level In COVID-19+ and Controls
Spadaro [64]	31	67 (55–75)	26/5	-	-	31 lower value than COVID-19- ARDS (*n* = 31)	P-Sel lower than ARDS controls, but higher than normal value

Abbreviation: P-sel = P-selectin; med= median; ICU = intensive care unit; (*n*) = number of patients in each class.

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
