# Peer review of "The Role of P-Selectin in COVID-19 Coagulopathy: An Updated Review"

_ijms, 2021, doi:10.3390/ijms22157942_

Round 1

Reviewer 1 Report

This review focuses on the changes of circulating p-selectin in patients with COVID-19. It is a comprehensive review. It would be better to give an meta analysis of the diagnostic and prognostic values of p-selection in COVID-19, particularly when the authors mentioned in ABSTRACT that p-selectin could serve as a marker of severity. In addition, a better hypothesis of how p-section is involved in the pathogenesis in COVID-19.

Author Response

Referee #1 (Comments to the Author):

This review focuses on the changes of circulating p-selectin in patients with COVID-19. It is a comprehensive review.

#1. It would be better to give a meta-analysis of the diagnostic and prognostic values of p-selection in COVID-19, particularly when the authors mentioned in ABSTRACT that p-selectin could serve as a marker of severity.

Author response: The need for a meta-analysis and statistical analysis of the published data on p-selectin in COVID-19 is well understood. However, because to the scarcity and inconsistency of reported evidence, the lack of standardized procedures for p-selectin testing and assessment, and the lack of validation in relevant patient populations, we decided not to go into a too complex and   unreliable analysis.

#2. In addition, a better hypothesis of how p-section is involved in the pathogenesis in COVID-19.

Author response:  We've added a sentence to the discussion to summarize the notion of how p-section is involved in COVID-19 pathogenesis, as suggested by the reviewer (line 315-317).

Reviewer 2 Report

Dear Sirs, congratulations on your review! The topic is in fact of high significance and further studies are desperately neded, thus, I believe, this  review will trigger other studies!

Please, do correct or explain the flow chart (29-11 does not equal 20), also there is a need to show a detailed search strategy as always in the systematic review; it may be addes as supplementary materials. 

If I may suggest anything else, try to omit "COVID-19 people"- that might be found somehow offensive, instead, simply use the word "patients"

Best regards

Author Response

Referee #2 (Comments to the Author):
Dear Sirs, congratulations on your review! The topic is in fact of high significance and further studies are desperately needed, thus, I believe, this review will trigger other studies!

#1.Please, do correct or explain the flow chart (29-11 does not equal 20), also there is a need to show a detailed search strategy as always in the systematic review; it may be added as supplementary materials. 

Author response: As suggested by the reviewer we have corrected the incorrect numbers in the flow chart. We understand the need of a thorough description of the search method in a systematic review.  When conducting a systematic search, the most crucial question to ask is if all of the possibly relevant articles were found using the search method. Given the short time span (one year) covered by the study, we are enough confident that all articles relating to P-selectin and COVID-19 have been reviewed and that the search approach utilized has been adequately stated in the text.

#2. If I may suggest anything else, try to omit "COVID-19 people"- that might be found somehow offensive, instead, simply use the word "patients"

Author response:  As suggested by the reviewer, we have changed the word “people” with “patients” (line 280).

This manuscript is a resubmission of an earlier submission. The following is a list of the peer review reports and author responses from that submission.